# Autocrine TGFβ1 Opposes Exogenous TGFβ1-Induced Cell Migration and Growth Arrest through Sustainment of a Feed-Forward Loop Involving MEK-ERK Signaling

**DOI:** 10.3390/cancers13061357

**Published:** 2021-03-17

**Authors:** Hendrik Ungefroren, Jessica Christl, Caroline Eiden, Ulrich F. Wellner, Hendrik Lehnert, Jens-Uwe Marquardt

**Affiliations:** 1First Department of Medicine, University Hospital Schleswig-Holstein, Campus Lübeck, D-23538 Lübeck, Germany; jessica.christl@student.uni-luebeck.de (J.C.); caroline.eiden@student.uni-luebeck.de (C.E.); Jens.Marquardt@uksh.de (J.-U.M.); 2Clinic for General Surgery, Visceral, Thoracic, Transplantation and Pediatric Surgery, University Hospital Schleswig-Holstein, Campus Kiel, D-24105 Kiel, Germany; 3Clinic for Surgery, University Hospital Schleswig-Holstein, Campus Lübeck, D-23538 Lübeck, Germany; ulrich.wellner@uksh.de; 4University of Salzburg, A-5020 Salzburg, Austria; hendrik.lehnert@sbg.ac.at

**Keywords:** transforming growth factor β, pancreatic cancer, breast cancer, cell growth, autocrine regulation, extracellular-regulated kinase, WAF1, SNAIL

## Abstract

**Simple Summary:**

Transforming growth factor (TGF) β signaling is intimately involved in nearly all aspects of tumor development and is known for its role as both a tumor suppressor in benign tissues and a tumor promoter in advanced cancers. This dual role is also reflected by cancer cell-produced TGFβ that eventually acts on the same cell(s) in an autocrine fashion. Recently, we observed that endogenous *TGFB1* can inhibit rather than stimulate cell motility in cell lines with high autocrine TGFβ production. The unexpected anti-migratory role prompted us to evaluate how autocrine TGFβ1 impacts the cells’ migratory and proliferative responses to exogenous (recombinant human) TGFβ. Surprisingly, endogenous *TGFB1* opposed the migratory and growth-inhibitory responses induced by exogenous TGFβ1 by driving a self-perpetuating feedforward loop involving MEK-ERK signaling. Our observation has implications for the use of TGFβ signaling inhibitors in cancer therapy.

**Abstract:**

Autocrine transforming growth factor β (aTGFβ) has been implicated in the regulation of cell invasion and growth of several malignant cancers such as pancreatic ductal adenocarcinoma (PDAC) or triple-negative breast cancer (TNBC). Recently, we observed that endogenous *TGFB1* can inhibit rather than stimulate cell motility in cell lines with high aTGFβ production and mutant KRAS, i.e., Panc1 (PDAC) and MDA-MB-231 (TNBC). The unexpected anti-migratory role prompted us to evaluate if aTGFβ1 may be able to antagonize the action of exogenous (recombinant human) TGFβ (rhTGFβ), a well-known promoter of cell motility and growth arrest in these cells. Surprisingly, RNA interference-mediated knockdown of the endogenous *TGFB1* sensitized genes involved in EMT and cell motility (i.e., *SNAI1*) to up-regulation by rhTGFβ1, which was associated with a more pronounced migratory response following rhTGFβ1 treatment. Ectopic expression of *TGFB1* decreased both basal and rhTGFβ1-induced migratory activities in MDA-MB-231 cells but had the opposite effect in Panc1 cells. Moreover, silencing *TGFB1* reduced basal proliferation and enhanced growth inhibition by rhTGFβ1 and induction of cyclin-dependent kinase inhibitor, p21^WAF1^. Finally, we show that aTGFβ1 promotes MEK-ERK signaling and vice versa to form a self-perpetuating feedforward loop that is sensitive to SB431542, an inhibitor of the TGFβ type I receptor, ALK5. Together, these data suggest that in transformed cells an ALK5-MEK-ERK-aTGFβ1 pathway opposes the promigratory and growth-arresting function of rhTGFβ1. This observation has profound translational implications for TGFβ signaling in cancer.

## 1. Introduction

Pancreatic ductal adenocarcinoma (PDAC) and triple-negative breast cancer (TNBC) are among the most aggressive and early metastasizing tumors [1,2]. Their high mortality is caused in part by the late diagnosis, which often only occurs in an advanced disease state, emphasizing the need for the identification of reliable biomarkers for early diagnosis or prognosis [3,4]. Both cancer types are highly heterogeneous diseases characterized by diverse molecular and morphological features with the quasi-mesenchymal/squamous subtype of human PDAC [5] or basal-like subtype of BC [6] having the worst prognosis of any of the recently identified subtypes. Their poor treatment response and early resistance against conventional treatments eventually leads to aggressive metastatic disease. Aberrantly activated signaling pathways in PDAC and TNBC such as that of transforming growth factor β (TGFβ) were identified as drivers of mesenchymal/squamous differentiation due to the ability to induce epithelial-mesenchymal transition (EMT). This complex genetic program confers migratory and invasive properties to epithelial cells during cancer, therefore, linking aberrant TGFβ signaling and EMT to PDAC and TNBC aggressiveness, loss of growth inhibition, and resistance to chemo- and radiotherapy [7,8,9]. Although TGFβ pathways have been extensively studied, the mechanisms leading to cancer promotion and development are still not completely understood. The predominant genomic alteration in PDAC, and to a lesser extent in TNBC, affects the *KRAS* gene [10,11]. Persistent Kirsten Rat Sarcoma (KRAS)-epidermal growth factor receptor (EGFR) pathway activation cooperates with TGFβ signaling to endow PDAC and TNBC tumor cells with chemoresistance, metastatic dissemination, and early recurrence [1,10,11,12].

In malignant but not benign cells, TGFβ1 has been shown to potently auto-induce its own expression [13,14], which in proximal tubular epithelial cells requires the coordinated, but independent positive regulation by SMAD3, p38, and extracellular signal-regulated kinase (ERK) signaling [15]. We recently employed two human cancer cell lines with high autocrine TGFβ1 (aTGFβ1) production, namely Panc1, a PDAC-derived line with a quasi-mesenchymal signature, and MDA-MB-231, a TNBC-derived line of the basal-like subtype, to elucidate the underlying signaling pathways. We were able to identify the small GTPase, Ras-related C3 botulinum toxin substrate 1B (RAC1B), a splice isoform of RAC1 and powerful inhibitor of rhTGFβ1-induced cell migration, as an upstream activator of *TGFB1* expression and TGFβ1 secretion [16]. In turn, aTGFβ1 induces SMAD3 protein expression [16] and basal p38 activation [17], suggesting their involvement in positive regulation of its own synthesis. However, whether aTGFβ1 also affects MEK-ERK signaling in PDAC and TNBC cells has not yet been analyzed.

It is generally believed that the ability to produce and secrete TGFβ that subsequently acts on the same cells or its neighbors in an autocrine or paracrine fashion can enhance a malignant phenotype [8]. However, a couple of observations suggest that endogenously produced aTGFβ and exogenous TGFβ can induce different signaling and target gene responses. For instance, endogenous TGFβ regulates the cell cycle through a pathway different from exogenous TGFβ with respect to sensitivity of effector proteins like cyclin-dependent kinase inhibitor 1 (p21^WAF1^) and CDK4 [18]. Moreover, previous work indicated that aTGFβ, rather than response to exogenous TGFβ, is an important protector against malignant progression. For instance, constitutively repressing endogenous TGFβ1 expression and aTGFβ activity in human colon carcinoma cells retained their functional receptor complexes and the ability to respond to exogenous TGFβ but led to a more progressed phenotype [19]. In order to abrogate aTGFβ signaling, the majority of studies have used either dominant-negative inhibition [20,21,22,23], reconstitution of the type II receptor (TβRII) [18,24], or inhibition of the activin receptor-like kinase 5 (ALK5) kinase activity [20,25,26,27] (for a comprehensive review see [28]). However, these approaches have important limitations for the following reasons: (i) they did not allow for a discrimination between the effects of the three different TGFβ isoforms, TGFβ1, 2, and 3, (ii) aTGFβ1 has been reported to be able to signal with respect to target gene expression, invasion but not proliferation in colon cancer cells that have lost the ability to produce functional TβRII as a result of microsatellite instability [25], and (iii) they are expected to alter the response to both exogenous and aTGFβ1 and thus preclude an assessment of how exogenous TGFβ1 interacts with aTGFβ1. This is a serious issue since in most studies, experiments were performed in medium with 10% fetal bovine serum (FBS), which may have contained high concentrations of latent or bioactive TGFβ1. In the present study, we, therefore, chose a more specific approach by targeting the ligand, TGFβ1, directly through RNAi-mediated knockdown of *TGFB1* to analyze how aTGFβ signaling impacts the chemokinetic and growth-inhibitory response to stimulation with exogenous, recombinant human TGFβ1 (rhTGFβ1). Our results provide evidence for the operation in cancer cells of an ALK5-MEK-ERK-aTGFβ1 pathway that opposes the promigratory and growth-arresting function of rhTGFβ1.

## 2. Results

### 2.1. Divergent Effects of Endogenous and Exogenous TGFβ1 on TGFβ Target Genes

Initially, we addressed the question of whether altering endogenous TGFβ1 levels would impact the response of EMT and invasion-associated genes to stimulation with exogenous rhTGFβ1. To this end, knocking down (KD) *TGFB1* in Panc1 cells by RNA_i_ interference (Panc1^TGFB1-KD^) significantly enhanced the stimulatory effect of rhTGFβ1 treatment on four invasion-promoting genes, *SNAI1*, *SNAI2*, *SERPINE1* (encoding plasminogen activator-inhibitor type I, PAI-1), and *F2RL1* (encoding proteinase-activated receptor 2, PAR2) (Figure 1A). Likewise, in MDA-MB-231^TGFB1-KD^ cells, *SNAI1*, *SNAI2*, *F2RL1* but not *SERPINE1* were more responsive to rhTGFβ1 (Figure 1B). For SNAIL, this effect was confirmed at the protein level in Panc1 cells (Figure 1C, left-hand blot). We also noted a more pronounced down-regulation of E-cadherin in Panc1^TGFB1-KD^ cells (Figure 1C, right-hand blot). Moreover, the number of spindle-shaped cells induced by treatment with rhTGFβ1 was greater in Panc1^TGFB1-KD^ than in control cells (Appendix A). Together, the observed changes in the expression of master EMT regulators and in cellular morphology strongly suggest that endogenous TGFβ1 antagonizes rhTGFβ1-induced EMT.

### 2.2. Knockdown of TGFB1 Enhances the Migratory Response to rhTGFβ1

Having shown that endogenous and rhTGFβ1 display antagonistic effects on several invasion-associated genes, we asked whether depleting cells of endogenous *TGFB1* expression would alter their sensitivity to rhTGFβ1-induced cell migration. Of note, under both basal conditions and in response to challenge with rhTGFβ1 MDA-MB-231^TGFB1-KD^ or Panc1^TGFB1-KD^ cells exhibited a dramatic increase in chemokinetic activity, which was particularly strong in Panc1 cells (Figure 2). Together, these data show that endogenous and exogenous rhTGFβ1 also exert antagonistic effects on cell migration and confirm the previous assumption that aTGFβ1 signaling may protect tumor cells from mesenchymal conversion and an associated increase in cell motility by non-autocrine, stromal cell-derived TGFβ1 [16].

### 2.3. Ectopic TGFB1 Antagonizes Basal and rhTGFβ1-Induced Migration in MDA-MB-231 Cells but Synergizes with rhTGFβ1-Induced Migration in Panc1 Cells

Given the somehow paradoxical nature of aTGFβ1 being anti-migratory, we wanted to know how ectopic (over)expression of TGFβ1 from a transfected TGFβ1-encoding expression vector (pTGFB1) in the same cells impacts cell migration. In MDA-MB-231 cells, ectopic expression of TGFβ1 inhibited both basal and rhTGFβ1-induced migratory activities (Figure 3A). Surprisingly, however, when introduced into Panc1 cells, the ectopic TGFβ1 on its own enhanced rather than suppressed migration over that of vector controls and when combined with rhTGFβ1 further enhanced its pro-migratory effect (Figure 3B). Moreover, forced expression of *TGFB1* synergized with RAC1B depletion in enhancing chemokinetic activity (Figure 3C, magenta curve/tracing D). Of note, enhanced migration of Panc1 cells in response to ectopic TGFβ1 expression was associated with increased activities of the *SNAI1* and *SNAI2* genes in Panc1 but not in MDA-MB-231 cells (Figure 3D). These results show that ectopically expressed TGFβ1 can either behave like aTGFβ1 and inhibit migration (in TNBC cells), or like rhTGFβ1 to stimulate migration (in PDAC cells). This is a significant observation given that both TGFβs are derived from the same coding sequence (albeit from genes of different structure and nuclear localization) and produced and secreted in an autocrine fashion.

### 2.4. Endogenous TGFB1 Opposes rhTGFβ1-Induced Growth Arrest and Induction of p21^WAF1^ Expression 

Next, we asked if endogenous TGFβ1 impacts the well-known growth arresting function of rhTGFβ1 on Panc1 [29] and MDA-MB-231 [30] cells. We transfected both cell lines with *TGFB1* siRNA and subsequently left the cells untreated or treated them for 50 h with rhTGFβ1 in normal growth medium. Results show that the silencing of endogenous *TGFB1* alone greatly decreased the number of cells (Figure 4A). Moreover, the growth-suppressing effect of rhTGFβ1 was more pronounced in TGFB1-KD cells compared to rhTGFβ1-treated control cells (Figure 4A). Since we failed to detect any changes in the number of apoptotic cells among the various treatment groups, we conclude that reduced proliferative activity accounted for the lower cell counts in untreated and rhTGFβ1-treated TGFB1-KD cells.

TGFβ1 is known to induce growth arrest by up-regulating the expression of the cyclin-dependent kinase inhibitor, p21^WAF1^, in pancreatic [29] and breast (Appendix A) epithelial cells. We, therefore, hypothesized that the reduced proliferative activity following *TGFB1* silencing might have been caused by derepression of p21^WAF1^. Strikingly, the abundance of p21^WAF1^ was enhanced in Panc1^TGFB1-KD^ or MDA-MB-231^TGFB1-KD^ cells vs. control transfectants and between non-rhTGFβ1-treated and rhTGFβ1-treated cells (Figure 4B). These data clearly show that aTGFβ1 opposes not only the pro-migratory effect of exogenous TGFβ1 but also its growth-suppressing function. Interfering with growth inhibition by exogenous TGFβ1 enhances mitotic activity, eventually resulting in a hyper-proliferative state that may have a role in early tumor development prior to mutational inactivation of Smad signaling.

### 2.5. Mutual Induction of aTGFβ1 or ERK Activation Sustains Proliferation

Prompted by the stimulating effect of aTGFβ1 on basal proliferation and its antagonism on rhTGFβ1-induced growth arrest in Panc1 and MDA-MB-231 cells, we next sought to elucidate the signaling events underlying TGFβ1 auto-induction. A previous study in fibroblasts has shown the involvement of ERK MAPK signaling in TGFβ1 mRNA transcription [15]. To reveal whether ERK signaling is required for basal and rhTGFβ1-induced expression of aTGFβ1 in pancreatic cancer cells, we measured its expression by qPCR and ELISA in Panc1 cells that have been treated, or not, with rhTGFβ1 in the presence or absence of U0126, a selective MEK1/2 inhibitor [31]. U0126 specifically targeted ERK signaling for inhibition in Panc1 (Appendix A) and MDA-MB-231 cells (Appendix A), and in Panc1 cells had no effect on C-terminal phosphorylation (= activation) of SMAD3 by rhTGFβ1 (Appendix A). Intriguingly, the abundance of *TGFB1* mRNA in both non-stimulated cells and rhTGFβ1-treated cells was dramatically reduced following inhibition of ERK activation (Figure 5A). Likewise, in U0126-treated cells the amount of TGFβ1 secreted into the culture medium was strongly decreased (Figure 5B).

Next, we focused on the role of the MEK-ERK pathway in auto-induction of TGFβ1. PDAC cells, i.e., Panc1, and TNBC MDA-MB-231 cells exhibit readily detectable levels of phosphorylated (p)ERK1/2 (Appendix A) as a result of oncogenic KRAS activity [11,32,33]. The activated ERK drives basal proliferation in these cells as evidenced by a dramatic decrease in DNA synthesis following inhibition of ERK activation with U0126 in Panc1 (Figure 5C) and MDA-MB-231 (Appendix A) cells. Since oncogenic KRAS^G12V^ and BRAF^V600E^ [34] have been reported to stimulate aTGFβ1 production and both the mitogenic function and the auto-induction of TGFβ1 converged on ERK signaling in a prostate carcinoma cell line [35], we considered the possibility that aTGFβ1 itself is involved in sustaining constitutive ERK activation. To this end, the levels of pERK1/2 in TGFB1-KD cells were found to be much lower than in control transfectants (Figure 5D). These data show that endogenous TGFβ1 and ERK signaling mutually enhance their expression/activation, eventually forming a self-perpetuating feedforward loop that drives basal proliferation and protect cells from the growth-suppressing effect of rhTGFβ1.

### 2.6. Effect of Inhibition of the ALK5 Kinase on TGFB1 Expression, Cell Migration, and ERK2 Activation

Previous studies have shown that TGFβ1 via ALK5 can activate ERK [36], an event that requires the ALK5 kinase and the adapter protein ShcA [37,38]. Moreover, treatment of MDA-MB-231 cells with SB431542 [30], a small molecule inhibitor of the ALK5 kinase [39], or of Panc1 cells with U0126 [40], increased the expression of p21^WAF1^. We, therefore, considered the possibility that SB431542 disrupts the ERK-TGFβ1 loop leading to a decrease in ERK activation and *TGFB1* expression and, as a consequence, to enhanced cell migration. Of note, the abundance of *TGFB1* mRNA in Panc1 cells (Figure 6A), or secreted TGFβ1 protein in MDA-MB-231 cells (Appendix A), was strongly reduced following SB431542 treatment, while that of the invasion-associated genes, *SNAI1* and *SERPINE1*, in Panc1 cells was increased (Figure 6A). As control, we exposed Panc1 cells to a combination of SB431542 and rhTGFβ1, which, as expected, alleviated (auto)induction of *TGFB1*, *SNAI1,* and *SERPINE1* by rhTGFβ1 (Figure 6A). Intriguingly, the SB431542 treatment also interfered with ERK activation (Figure 6B). Given the decrease in both *TGFB1* mRNA and steady-state pERK levels and the increases in *SNAI1* and *SERPINE1* expression, we reasoned that this should impact the cells’ propensity for cell migration. To this end, treatment of Panc1 cells with SB431542 but not PP2, a Src family kinase inhibitor that blocks TGFβ/Smad and p38 MAPK signaling in a Src-unrelated fashion [41], enhanced their chemokinetic activity (Figure 6C). Likewise, both Panc1^TGFB1-KD^ and control cells responded to SB431542 treatment with elevated migration (Figure 6D). The promigratory effect of SB431542 was duplicated in MDA-MB-231 cells, while PP2 or the chemically related p38 MAPK inhibitor SB203580 had no effect (Appendix A). Finally, we found that SB431542 but not SB203580 [42], decreased cell cycle progression as revealed by (^3^H)-thymidine incorporation (Figure 6E). The failure of siRNA-mediated knockdown of either mothers against decapentaplegic homolog 2 (SMAD2) or SMAD3 to decrease DNA synthesis (Figure 6F) revealed that the antiproliferative effect of SB431542 was not due to inhibition of the Smad-activating function of the ALK5 kinase. From these data, we conclude that *TGFB1* expression, aTGFβ1-mediated inhibition of cell invasion, and promotion of growth is driven at least in part by the ability of ALK5 to induce MEK-ERK signaling.

**Figure 6 cancers-13-01357-f006:**
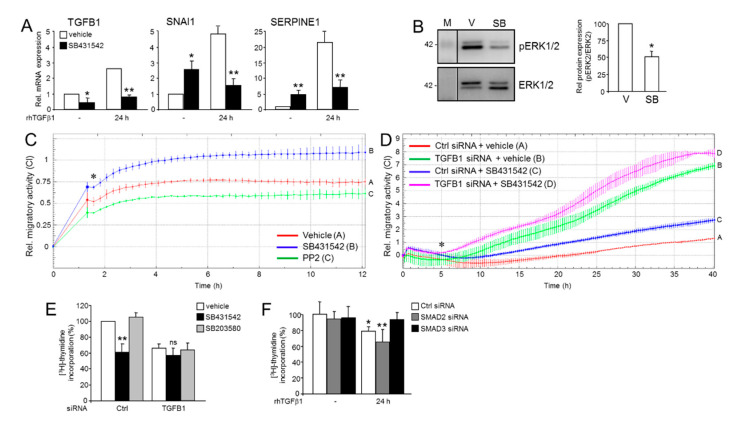
Effects of SB431542 on endogenous TGFβ1 expression, ERK activation, and cell migration. (**A**) Panc1 cells were treated, or not, for 24 h in the absence or presence of SB431542 (1 µM), or vehicle (0.1% DMSO) followed by qPCR analysis of *TGFB1*, *SNAI1* or *SERPINE1* and GAPDH and TBP as internal control. Data represent the normalized mean ± SD of triplicate wells from a representative experiment. The asterisks indicate significance relative to the vehicle control. (**B**) Panc1 cells were treated, or not, for 24 h with 1 µM SB431542 or vehicle (V) followed by immunoblotting for pERK1/2 and ERK1/2 as loading control. The graph underneath the blots shows quantitative data (mean ± SD from three independent assays). (**C**) Panc1 cells were subjected to real-time cell migration assay in the presence or absence of SB431542 (1 µM) or PP2 (10 µM). Mean ± SD of triplicate wells. Differences between the red curve (tracing A) and the blue curve (tracing B) were first significant at 1:40 (∗) and all later time points. (**D**) Panc1^TGFB1-KD^ and control (ctrl) cells were analyzed as in (C) in the presence or absence of SB431542 (1 µM). Mean ± SD of triplicate wells. Differences between the green curve (tracing B) and the magenta curve (tracing D) were first significant at 5:00 (∗) and all later time points. (**E**) Panc1 cells were treated, or not, for 24 h with the indicated concentrations of SB431542 or vehicle (V) followed by (^3^H)-thymidine incorporation. (**F**) Panc1 cells were transfected with siRNA to either SMAD2 or SMAD3, or a scrambled control (ctrl) siRNA, followed by (^3^H)-thymidine incorporation. Successful knockdown of SMAD2 and SMAD3 was verified by immunoblot analysis [43] and, functionally, by the ability of SMAD3 siRNA to block growth inhibition by rhTGFβ1 [44]. Data in (**E**,**F**) are the mean ± SD from 6 wells processed in parallel and are representative of 5 and 3 experiments, respectively. The asterisks (∗) indicate significance compared to vehicle control; * *p* < 0.05; **, *p* < 0.01; ns, not statistically significant.

## 3. Discussion

Earlier experiments revealed that inhibition of aTGFβ1 in pancreatic and BC cells with known aTGFβ1 production, surprisingly, enhanced the cells’ migratory activity and reduced basal proliferation [16], thus exhibiting effects antagonistic to those of rhTGFβ1. This prompted us to analyze how modulating endogenous TGFβ1 expression impacts the cells’ response to treatment with exogenous (rh)TGFβ1. In the present study, we observed that silencing the endogenous *TGFB1* gene in Panc1 or MDA-MB-231 cells reduced cell counts in the absence of exogenously applied rhTGFβ1, while strongly enhancing the stimulatory effect of exogenous/rhTGFβ1 on both invasion and growth inhibition. This resembled the situation with RAC1B, consistent with its role as an upstream activator of endogenous *TGFB1* and TGFβ1 secretion [16] and potent antagonist of rhTGFβ1-induced invasion and growth arrest [43].

We further observed that ectopically expressed TGFβ1 can behave as either endogenous or exogenous TGFβ1, depending on the cancer type. Intriguingly, the strong increase in rhTGFβ1-dependent migratory activity following blockage of aTGFβ synthesis/secretion was associated with a more pronounced induction by rhTGFβ1 of EMT/invasion-associated gene expression and the cell cycle inhibitor, p21^WAF1^. As speculated earlier, low concentrations of aTGFβ1 may be able to desensitize the pathway and block the action of exogenous TGFβ1. It has, indeed, been reported that exposure of cells to low and high concentrations of TGFβ may have different and even opposing outcomes on cell migration [45].

The signaling events underlying the antagonism between endogenous and exogenous TGFβ1 nevertheless remains elusive but do not appear to involve quantitative changes in the expression of central TGFβ signaling intermediates such as ALK5 or SMAD7 expression [16]. Rather, we identified the MEK-ERK signaling pathway as a driver of both the mitogenic action of aTGFβ1 and endogenous *TGFB1* expression under basal conditions and following stimulation with rhTGFβ1. This is consistent with reduced levels of DNA synthesis and pERK1/2 following either *TGFB1* silencing, or pharmacologic MEK inhibition as control (Figure 5). Conversely, blocking ERK activation mitigated both basal and rhTGFβ1-induced *TGFB1* expression. In fact, it appears that aTGFβ1 and MEK-ERK signaling mutually sustain their expression/activation to form a self-perpetuating feedforward loop that opposes the actions of exogenous TGFβ1 (Figure 7). Given the transcriptional up-regulation of *TGFB1* by rhTGFβ1 and the functional antagonism between aTGFβ1 and rhTGFβ1 on cell motility and proliferation [16], this resembles the situation with the inhibitory Smad, SMAD7, which also acts as an endogenous albeit intracellular inhibitor of TGFβ signaling that provides feedback inhibition [46]. Interestingly, abrogation of aTGFβ signaling via dominant-negative interference with TβRII or kinase inhibition of the type I receptor (TβRI) ALK5 in a BC cell line decreased the levels of activated ERK but increased those of p21^WAF1^ and induction of apoptosis [20]. Moreover, U0126-mediated inhibition of ERK activation has been shown by others to sensitize Panc1 cells to rhTGFβ1-induced up-regulation of p21^WAF1^ [40], together suggesting that ERK opposes growth-suppressive TGFβ signals and promotes proliferation in the cancer state. This appears to be a distinguishing feature of transformed cells since during carcinogenesis, pERK initially facilitates and later antagonizes exogenous TGFβ-mediated cell cycle arrest [40].

A growth-promoting effect of aTGFβ1 has been described previously in MDA-MB-231 cells [47] and in colon carcinoma cells [18,19] with aTGFβ regulating the cell cycle through a pathway different from exogenous TGFβ with respect to sensitivity of p21^WAF1^ and CDK4 [18]. In a single prostate carcinoma cell line, the mitogenic function of TGFβ1 was dependent on ERK signaling [35]. However, the authors of this study have used rec. porcine TGFβ1 to stimulate their cells and to mimic the effect of aTGFβ. Because of this variation and the lack of data showing that this cell line was indeed capable of TGFβ1 auto-production, the data were not directly comparable with ours. Nevertheless, the authors provided evidence that mitogenic conversion of aTGFβ1 is dependent on oncogenic RAS proteins. Of note, proliferation and ERK signaling in both PDAC and TNBC-derived cells is driven by mutant KRAS, albeit different mutations, via activation of CRAF1, and both CRAF1 and BRAF^V600E^ [34] can induce TGFβ1 secretion. Moreover, activation of Raf in MDCK cells was able to block the ability of rhTGFβ to induce apoptosis [12], which is in good agreement with our data showing that aTGFβ interferes with rhTGFβ-induced growth inhibition and migration.

Another interesting issue relates to the question of whether MEK-ERK signaling and aTGFβ1 production can also be triggered by wild-type RAS proteins and, if so, what the upstream activator(s) are. In Panc1 cells, further ERK activation is induced by mitogenic stimuli [33] due to activation of KRAS protein encoded by the wild-type allele. A likely candidate is EGF, which is a strong inducer of MEK-ERK signaling in Panc1 (Appendix A) and other cell types and may be able to target this newly identified pro-proliferative circuit of aTGFβ1-ERK to antagonize growth arrest by exogenous TGFβ1 (Figure 7). Interestingly, EGF has been shown to abrogate the antiproliferative effects of rhTGFβ in primary human ovarian cancer cells, representing a potential non-mutational mechanism in cells lacking mutations in SMAD4, or the receptors, to inhibit exogenous TGFβ signaling and contributing to uncontrolled proliferation [48]. Likewise, in mammary epithelial cells, expression of mutant HER2 or HRAS^G12V^ activated aTGFβ1 expression and signaling through a mechanism involving activation of RAC1 [49]. Here, we pursued the idea of direct ERK activation by the TGFβ receptor(s) based on the realization that ALK5 is a dual-specificity kinase, with its tyrosine kinase function being able to directly phosphorylate ShcA [37,38]. Tyrosine phosphorylation of ShcA by ALK5 is dependent on the kinase domain and is inhibitable by SB431542 [37]. Intriguingly, blocking the ALK5 kinase with SB431542 in Panc1 or MDA-MB-231 cells decreased the abundance of *TGFB1* mRNA or secreted TGFβ1 protein and pERK and enhanced invasive gene expression and chemokinesis, thus mimicking the inductive effect of *TGFB1* silencing on cell motility. Moreover, treatment with SB431542 reduced the basal proliferation of Panc1 cells in a Smad and p38 MAPK-independent manner. Together with the observation that SB431542 inhibited cell growth with up-regulation of p21^WAF1^ expression in MDA-MB-231 and other non-BC cell lines [30], we arrived at the conclusion that ALK5-mediated activation of MEK-ERK signaling accounts, at least in part, for fueling and sustaining the ERK-aTGFβ1 regulatory loop (Figure 7). Another observation by Koo and colleagues [30], namely, that treatment with SB431542 resulted in down-regulation of the transcription factor Sp1 together with the notion that *TGFB1* but not *TGFB2* is transcriptionally regulated by Sp1 [50] provided a possible mechanistic explanation of how the ALK5-MEK-ERK non-canonical pathway drives aTGFβ1 synthesis.

Further studies are underway to clarify if ShcA represents an upstream driver of this process. Silencing ShcA expression in non-transformed cells (NMuMG, HaCaT) induced EMT, cell migration, invasion, and dissemination, which was dependent upon aTGF-β signaling. However, rather than modulating EMT through the Erk pathway ShcA acted through suppressing Smad3 activation by competing with Smad3 for binding to ALK5 [38]. It, therefore, appears that during (Ras-induced) transformation, cells have switched the coupling of aTGFβ production from Smad to ERK signaling, paralleling the mitogenic switch of ERK [40].

Differential responsiveness of cells to autocrine and exogenous TGFβ1 has been described with both exhibiting differences in signaling, particularly in their requirements for the different receptor types. Specifically, functional TβRII has been shown to be dispensable for autocrine but not exogenous TGFβ1 [25], representing a potential mechanism to separate cellular responses to TGFβ1 from autocrine sources and from paracrine sources. Previous findings from our laboratory indicated that *TGFB1* silencing did not alter the abundance of SMAD7 mRNA or ALK5 protein expression [16]. An elucidation of the different receptors and co-receptors utilized by endogenous and exogenous TGFβ, their subcellular localization, and state of activation seems to be key to better understand the functional antagonism.

Interestingly, *TGFB1* silencing allowed for enhanced induction by rhTGFβ1 of other prominent target genes, i.e., *SNAI1* and *WAF1*, providing a molecular framework for the inhibitory effect of aTGFβ1 on rhTGFβ1-induced invasion and growth arrest. In a prostate cancer cell line, where mitogenic conversion of TGFβ1 required oncogenic HRAS^G12V^, p21^WAF1^ has been identified as a potential executor of this program [35]. Interestingly, both Panc1 and MDA-MB-231 cells contain gain-of-function mutations in KRAS (G12V and G13D, respectively) that are known to drive constitutive ERK signaling [51] and suppress p21^WAF1^ in these cells. Moreover, induction of *SNAI1* by rhTGFβ1 in Panc1 cells, which was greatly enhanced in TGFB1-KD cells (Figure 1B), is known to repress proliferation [52] and to be highly dependent on cooperation with active KRAS [32] and on MEK-ERK signaling [53].

An intriguing observation was that although both autocrine and ectopically expressed TGFβ1 were expressed and secreted by the same cells, they both inhibited cell invasion in MDA-MB-231 cells but displayed antagonistic effects in Panc1 cells. An explanation of why in pancreatic cells ectopic TGFβ1 behaves like exogenous TGFβ1 is not readily available but we are currently trying to decipher whether differences in total expression or conversion from latent to bioactive TGFβ1 account for this.

The finding that aTGFβ1 impairs invasive activities induced by exogenous/rhTGFβ1 suggests an anti-oncogenic function in late-stage carcinomas when malignant progression is largely driven by high concentrations of stromal cell-derived TGFβ1 in the tumor microenvironment. However, the ability of aTGFβ1 to interfere with the growth-arresting and, hence, tumor-suppressive function of (exogenous) TGFβ1 also makes it a potential oncogenic driver in early PDAC and BC development. Interestingly, murine pre-neoplastic pancreatic epithelial cells with mutant KRAS transiently exposed to exogenous TGFβ1, i.e., corresponding in vivo to pulses of TGFβ1 produced during chronic pancreatitis, a known risk factor for PDAC [54], converted to a partially mesenchymal (PM), progenitor-like, and hyper-proliferative state in vitro, which was stable and maintained by aTGFβ [27]. These PM cells, like Panc1, shared molecular and phenotypic features with the quasi-mesenchymal subtype of human PDAC and in vivo formed ductal structures resembling human PanINs. Unfortunately, a mechanistic explanation for the hyper-proliferation was not supplied in this study, but it is conceivable that it resulted—at least in part—from aTGFβ1 blocking the growth-inhibitory effect of exogenous TGFβ1. It is tempting to speculate that a finely tuned balance of the opposing actions of aTGFβ and exogenous TGFβ1 does not only control invasion and proliferation, but by inducing a partial/hybrid EMT also generates cancer stem cells and promotes resistance to anti-cancer drugs [55,56].

## 4. Materials and Methods

### 4.1. Cell Lines and Treatments

The PDAC-derived cell line, Panc1, and the TNBC-derived cell line, MDA-MB-231, were propagated in RPMI 1640 basal medium supplemented with 10% FBS, 1% Penicillin-Streptomycin-Glutamine (PSG, Life Technologies, Darmstadt, Germany) and 1% sodium pyruvate (Merck Millipore, Darmstadt, Germany). In some experiments, cells were stimulated with 5 or 10 ng/mL of human either rhTGFβ1 (#300-023, ReliaTech, Wolfenbüttel, Germany) or EGF (PeproTech, Hamburg, Germany). The MEK inhibitor U0126, the Src family kinase inhibitor PP2, the p38 MAPK inhibitor SB203580, and the Rac1 inhibitor NSC23766 [57] were purchased from Calbiochem/Merck (Darmstadt, Germany), and the ALK5 inhibitor SB431542 from Sigma (Deisenhofen, Germany).

### 4.2. Transient Transfections

For transient transfection, cells were seeded on day 1 into 12-well plates (Nunc, Roskilde, Denmark) and transfected twice, on days 2 and 3, serum-free with either 25 or 50 nM of prevalidated siRNAs specific for RAC1B or the respective scrambled controls. The TGFB1 siRNA (#1027416, a mixture of four different pre-evaluated siRNAs) was provided by Dharmacon (Lafayette, CO, USA) and the SMAD2 and SMAD3 siRNAs from Qiagen (Hilden, Germany). An expression vector for full-length human *TGFβ1 (*#SC119746) was purchased from OriGene Technologies Inc. (Rockville, MD, USA). SiRNAs or plasmids were transfected into cells serum-free for 4 h using Lipofectamine 2000 (Life Technologies) according to the manufacturer’s recommendations and previous descriptions [16,17,43].

### 4.3. Quantitative Real-Time PCR Analysis

Total RNA was isolated from Panc1 or MDA-MB-231 cells with the RNeasy Kit (Qiagen, Hilden, Germany) according to manufacturer’s instructions. For each sample, 2.5 μg RNA was reverse transcribed with 200 U of M-MLV Reverse Transcriptase and 2.5 μM random hexamers (1 h, 37 °C). Target gene mRNA expression was quantified by qPCR on an I-Cycler (BioRad, Munich, Germany) with Maxima SYBR Green Master Mix (Thermo Fisher Scientific, Schwerte, Germany) and normalized to the expression of either TBP or GAPDH. PCR primer sequences are provided in Appendix A.

### 4.4. Western Blotting

Our Western blotting procedure was described in detail earlier [16,17,43]. Total protein concentrations were determined with the DC Protein Assay (BioRad). Proteins were fractionated by PAGE on mini-PROTEAN TGX any-kD precast gels and blotted onto PVDF membranes. The primary antibodies included anti-HSP90 (Santa Cruz Biotechnology, Heidelberg, Germany, #sc-13119), anti-Rac1b (Merck Millipore, Darmstadt, Germany, #09-271), anti-E-cadherin and anti-Cip1/WAF1 (BD Transduction Laboratories, Heidelberg, Germany, #610181 and #610233, respectively), anti-Snail and anti-phospho-ERK1/2 (Cell Signaling Technology, Frankfurt/Main, Germany, #4719 and #4370, respectively), anti-ERK1/2 (R&D Systems, Wiesbaden, Germany, #AF1576), anti-GAPDH (14C10, Cell Signaling Technology, #2118), and anti-β-actin (Sigma). Incubation with HRP-linked secondary antibodies (Cell Signaling Technology, anti-rabbit, #7074, and anti-mouse, #7076) was followed by chemoluminescent detection of proteins on a ChemiDoc XRS+ System with Image Lab Software (BioRad) using Amersham ECL Prime Detection Reagent (GE Healthcare, Munich, Germany). The signals for the proteins of interest were normalized to bands for the housekeeping genes GAPDH or HSP90.

### 4.5. TGFβ1 ELISA

ELISA-based measurements of TGFβ1 were performed as described in detail earlier [16,17] using the TGFβ1-specific ELISA (Human/Mouse TGF beta1 ELISA Ready-SET-Go!) from eBioscience/Affymetrix Inc. (San Diego, CA, USA) with the only modification that total rather than bioactive TGFβ1 was measured.

### 4.6. (^3^H)-Thymidine Incorporation Assay

Labeling of the cells with methyl (^3^H)-thymidine was essentially done as outlined in detailed earlier with minor modifications [29].

### 4.7. Real-Time Cell Migration Assays

Migratory activities of Panc1 and MDA-MB-231 cells were determined with xCELLigence^®^ technology (Agilent Technologies, Santa Clara, CA, USA, supplied by OLS, Bremen, Germany) as outlined in detail in the instruction manual and previous publications [16,17,43] except for some modifications. The lower side of the membrane of the CIM-Plate 16 (https://www.agilent.com/en/product/cell-analysis/real-time-cell-analysis/rtca-microplates/rtca-cim-plates-741221#productdetails (accessed on 2 March 2021)) was coated with 30 μL of a 1:1 (*v*/*v*) mixture of collagens I and IV to facilitate adherence of the cells and to enhance signal intensities. A total of 60,000 cells transfected with either ctrl or TGFB1 siRNA and resuspended in serum-reduced (1% FBS) culture medium were loaded per well, and in some experiments, half of each transfectant received rhTGFβ1 (5 ng/mL) or vehicle.

### 4.8. Proliferation Assays

Panc1 or MDA-MB-231 cells were seeded at 100,000 cells per 12-well and transfected the next day and the day after with 50 nM each of ctrl siRNA or TGFB1 siRNA as outlined above. Four hours after the second round of transfection, cells were detached and seeded at 50,000 cells per 6 well. Following reattachment (16 h later), cells were stimulated with rhTGFβ1 (Panc1: 5 ng/mL, MDA-MB-231: 10 ng/mL) for 50 h, lifted by trypsinization and counted using the Cedex XS device (Roche Diagnostics, Mannheim, Germany).

### 4.9. Statistical Analysis

Statistical significance was calculated using either the unpaired two-tailed Student’s *t*-test or the Wilcoxon-test. Results were deemed significant at *p* < 0.05 (denoted by one asterisk). For some data, higher levels of significance were calculated and denoted by two or three asterisks (*p* < 0.01 or *p* < 0.001, respectively).

## 5. Conclusions

Given the current concept of autocrine TGFβ as a driver of tumor progression, our observation that endogenous autocrine TGFβ1 can also block cell motility was surprising and provoked the question how modulating expression of *TGFB1* impacts the migratory and proliferative responses to exogenous/recombinant human TGFβ1. Surprisingly, silencing endogenous *TGFB1* in PDAC and TNBC-derived cancer cells with known autocrine TGFβ1 production allowed exogenous TGFβ1 to elicit a more pronounced migratory and growth-inhibitory response. This can be interpreted to mean that in vivo cancer cells utilize autocrine TGFβ1 to protect themselves against the actions of stromal cell-derived paracrine TGFβ and suggest the possibility that a finely tuned balance of the antagonistic actions of autocrine and exogenous TGFβ(1) also controls the generation of EMT phenotypes with enhanced plasticity and stem cell potential. Last but not least, our data challenge the view that autocrine TGFβ production is always a feature of the “dark side” of TGFβ in cancer progression.

## Figures and Tables

**Figure 1 cancers-13-01357-f001:**
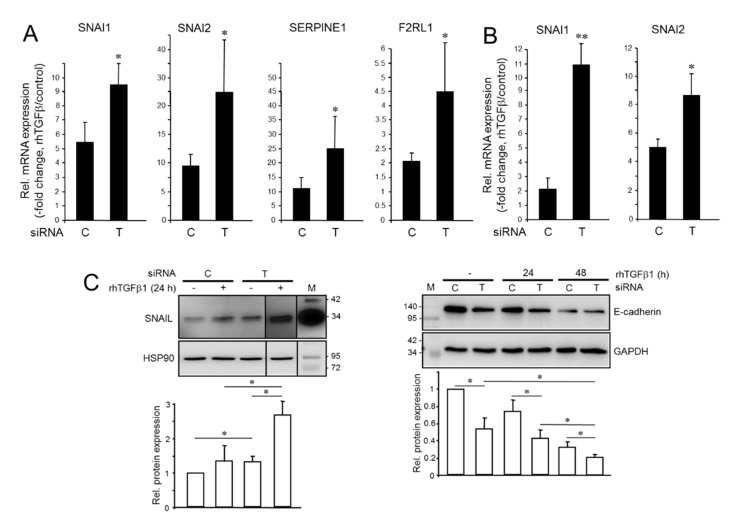
Effect of endogenous and recombinant human transforming growth factor β1 (rhTGFβ1) on gene expression in tumor cells with aTGFβ production. (**A**) Panc1^TGFB1-KD^ cells were transiently transfected twice on 2 consecutive days with 50 nM each of siRNA directed against TGFB1 (T) or a scrambled control (C) siRNA and incubated for another 48 h. Cells were then treated with rhTGFβ1 for 24 h and analyzed by qPCR for expression of the indicated genes and TATA box-binding protein (TBP) as a reference gene. Data are displayed as fold induction by rhTGFβ1 treatment over non-treated controls (means ± SD; *n* = 3 (SNAI1, SERPINE1), *n* = 4 (SNAI2, F2RL1)). (**B**) As in (**A**) except that MDA-MB-231^TGFB1-KD^ were analyzed. (**C**) Western blot analysis of SNAIL (left-hand blot) and E-cadherin (right-hand blot) in Panc1^TGFB1-KD^ cells treated, or not, with rhTGFβ1. Detection of heat shock protein 90 (HSP90) or glyceraldehyde 3-phosphate dehydrogenase (GAPDH) served as a loading control. The graphs below the blots show data quantification from densitometric readings of band intensities (mean ± SD, *n* = 3). The asterisks denote significance. The vertical lines between lanes 3, 4, and 5 of the left blot indicate that irrelevant lanes have been removed. Successful knockdown of *TGFB1* was verified by ELISA of secreted TGFβ1 in culture supernatants (Appendix A). M, molecular weight marker. Numbers to the right or left of the blots denote band sizes in kDa. *, *p* < 0.05; **, *p* < 0.01.

**Figure 2 cancers-13-01357-f002:**
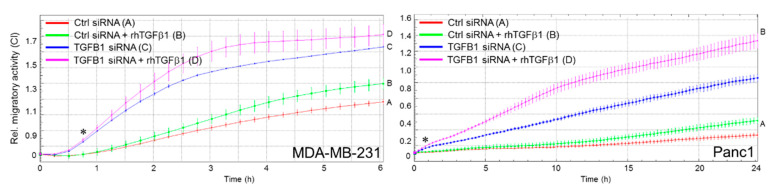
Effect of knockdown of endogenous *TGFB1* on basal and rhTGFβ1-dependent migration in triple-negative breast cancer (TNBC) and pancreatic ductal adenocarcinoma (PDAC)-derived tumor cells. MDA-MB-231 (left-hand graph) or Panc1 (right-hand graph) cells were transiently transfected with 50 nM of either a scrambled control (Ctrl) siRNA or TGFB1 siRNA and 48 h later subjected to impedance-based real-time measurement of random cell migration in the presence of rhTGFβ1 (5 ng/mL) for 24 h. Data represent the mean ± SD of 3–4 parallel wells for each condition. Differences between the green curves (tracing B) and the magenta curves (tracing D) were first significant at the 0:45 h time point (∗) and remained so during the entire observation period. Statistical significance was determined with the unpaired two-tailed Student’s *t*-test from the raw data of the various curves. Successful knockdown of *TGFB1* was verified by ELISA of secreted TGFβ1 in culture supernatants.

**Figure 3 cancers-13-01357-f003:**
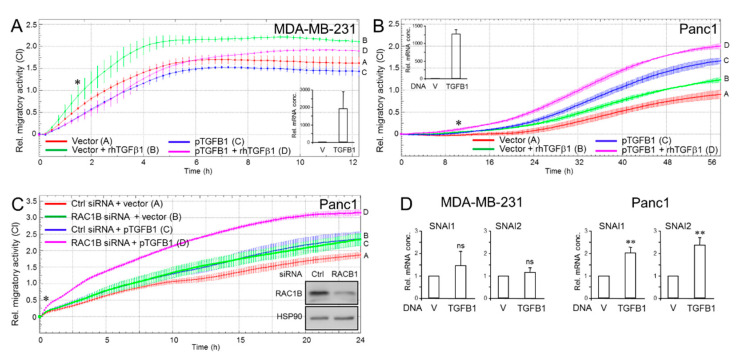
Effect of ectopic overexpression of *TGFB1* in MDA-MB-231 and Panc1 cells on cell motility. (**A**) MDA-MB-231 cells were transfected with either empty vector or an expression vector encoding TGFβ1 (pTGFB1) and 48 h later subjected to real-time measurement of random cell migration in the absence or presence of rhTGFβ1 (5 ng/mL). Differences between the green curve (tracing B) and the magenta curve (tracing D) were first significant at 1:30 (∗) and all later time points. (**B**) As in (A), except that Panc1 cells ectopically overexpressing TGFβ1 cells were assayed. Differences between the green curve (tracing B) and the magenta curve (tracing D) were first significant at 10:24 (∗) and all later time points. Overexpression of transfected TGFβ1 in (**A**,**B**) was validated by qPCR analysis (insets), and by ELISA with culture supernatants from pTGFB1 transfected cells with or without the addition of rhTGFβ1 (Appendix A). (**C**) Panc1 cells were transiently cotransfected with Ctrl siRNA or RAC1B siRNA along with either empty vector or pTGFB1. Then, 48 h later the transfectants were analyzed on the xCELLigence platform for changes in migratory activities. Differences between the green curve (tracing B) and the magenta curve (tracing D) were first significant at 1:00 (∗) and all later time points. Successful knockdown of RAC1B was verified by immunoblotting with HSP90 as loading control (inset). Data in (**A**–**C**) are the mean ± SD from 3 or 4 parallel wells. (**D**) QPCR analysis *SNAI1* and *SNAI2* in MDA-MB-231 and Panc1 cells (mean ± SD, *n* = 3). V, empty vector; TGFB1, TGFβ1 expression vector. Ns, not significant. *, *p* < 0.05; **, *p* < 0.01.

**Figure 4 cancers-13-01357-f004:**
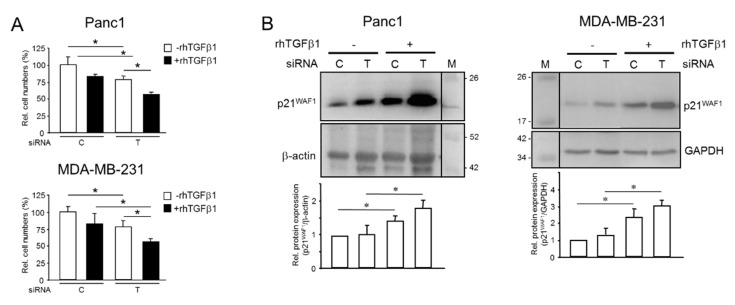
Effect of silencing *TGFB1* on rhTGFβ1-induced growth arrest in pancreatic and breast cancer cells with aTGFβ production. (**A**) Panc1 or MDA-MB-231 cells were transiently transfected with 50 nM of either control (C) siRNA or TGFB1 (T) siRNA and subsequently subjected to treatment with rhTGFβ1 (5 ng/mL) for 50 h. Cells were trypsinized and counted. Data (cell numbers) shown are representative of three assays, with non-rhTGFβ-treated control transfectants set arbitrarily at 100 (mean ± SD, *n* = 3). (**B**) Immunoblot analysis of p21^WAF1^ in Panc1 or MDA-MB-231 cells transfected as outlined in (**A**) but treated with rhTGFβ1 for 24 h. The graphs underneath the blots show results from densitometry-based quantification of band intensities (mean ± SD, *n* = 3). Successful knockdown of *TGFB1* in (**A**,**B**) was verified by ELISA of secreted TGFβ1. The asterisks (∗) indicate significance (*p* < 0.05).

**Figure 5 cancers-13-01357-f005:**
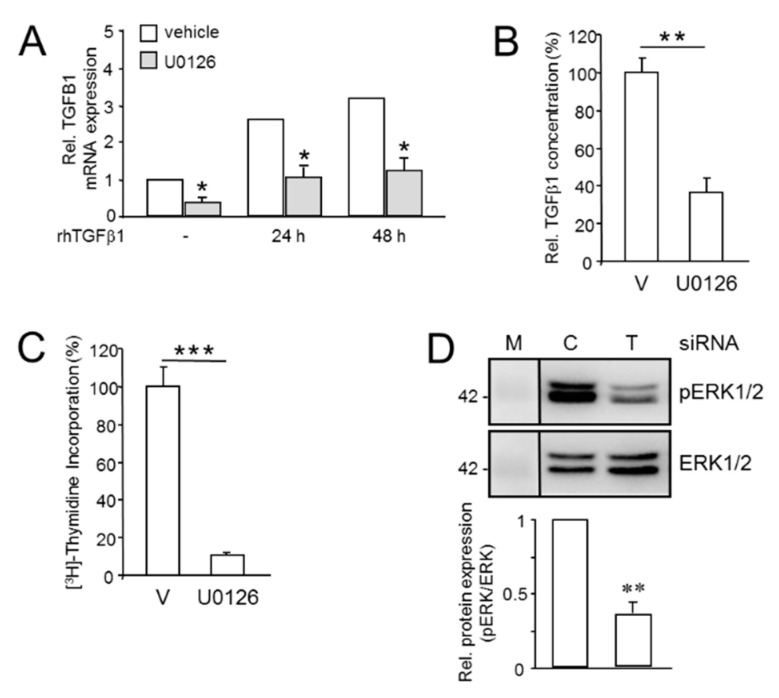
Mutual regulatory interactions between aTGFβ1 and ERK signaling enhances proliferation. (**A**) Panc1 cells were treated, or not (-), for 24 or 48 h with rhTGFβ1 (5 ng/mL) in the absence or presence of U0126 (10 µM) or vehicle (0.1% dimethyl sulfoxide, DMSO) followed by qPCR analysis of TGFB1 and GAPDH and TBP as internal control. Data represent the normalized mean ± SD of triplicate wells from a representative experiment. The asterisks indicate significance relative to the vehicle ctrl. (**B**) As in (**A**), except that cells were switched to serum-reduced medium (0.5% FBS) prior to U0126 treatment. Cells were allowed to condition their growth media for 24 h. Aliquots of conditioned media were subjected to ELISA measurement of total (bioactive + latent) TGFβ1. Data are the mean ± SD of triplicate samples. (**C**) Panc1 cells were treated, for 24 h with U0126 (25 µM) or vehicle (V) followed by (^3^H)-thymidine incorporation. Data shown are the mean ± SD from 6 wells processed in parallel and are representative of 3 experiments. The asterisks indicate significance. (**D**) Panc1 cells were transfected with control or TGFB1 siRNA as described in the legend to Figure 1, 24 h later stimulated with rhTGFβ1 for 1 h and subjected to immunoblotting for phospho-ERK1/2 (pERK1/2), and total ERK1/2 as a loading control. Data represent the mean ± SD from three independent assays. Successful knockdown of *TGFB1* was verified by ELISA of secreted TGFβ1. The asterisks (∗) indicate significant differences relative to the respective controls. *, *p* < 0.05; **, *p* < 0.01; ***, *p* < 0.001.

**Figure 7 cancers-13-01357-f007:**
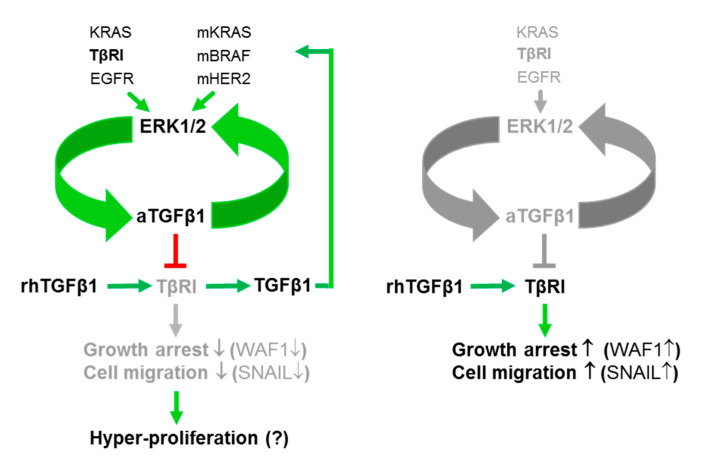
Cartoon illustrating the interactions of exogenous and endogenous/aTGFβ that operate in PDAC- and TNBC-derived tumor cells. Left-hand side, the aTGFβ1 forms a regulatory feedforward loop with ERK1/2 to sustain high-level ERK activation. This circuit prevents exogenous TGFβ1 (rhTGFβ1 in vitro or paracrine and stromal cell derived-TGFβ1 in vivo) from inducing growth arrest or cell migration through TβR1/ALK5 via induction of p21^WAF1^ (WAF1) or SNAIL, respectively. The aTGFβ1-ERK loop is driven by mutant (m) and wild-type versions of RAS, RAF, or EGFR/Erb-b2 receptor tyrosine kinase 2 (HER2) and additionally through TβRI/ALK5. Exogenous TGFβ1 also transcriptionally up-regulates *TGFB1,* and the resulting TGFβ1 protein via mutant RAS-ERK signaling can provide feedback inhibition of its own production. Right-hand side, in normal/benign cells, the aTGFβ1-ERK autoregulatory loop is non-functional due to the lack of mutant RAS/RAF proteins or lower expression or activation of TβRI/ALK5, but low-level activation of the aTGFβ1-ERK circuit is eventually achieved through EGFR activation via (wild-type) RAS. Green arrows denote activation or induction and red lines inhibition. Grey-shaded arrows/lines indicate inactivation.

## Data Availability

Data are contained within the article or Appendix A.

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
