# Peer review of "Autocrine TGFβ1 Opposes Exogenous TGFβ1-Induced Cell Migration and Growth Arrest through Sustainment of a Feed-Forward Loop Involving MEK-ERK Signaling"

_cancers, 2021, doi:10.3390/cancers13061357_

Round 1
Reviewer 1 Report
Ungefroren et al described the existence of an autocrine-backed feed-forward loop involving MEK-ERK opposing exogenous TGFβ1. Although performed exclusively in vitro, the work is well designed. However, some points need to be improved.
1. First of all, Results and Discussion must be better characterized. In particular, the Results section should be simplified by moving into Discussion comments involving literature data, that often make it difficult understanding the results shown.
2. Fig. 1C: it is not clear why the last band on the right was added: is it part of a different experiment? In a representative WB the bands should belong to treated and untreated samples from the same experiment and migrated to the same gel.
3. Fig 2: the Ctrl siRNA and TGFB1 siRNA conditions constitute the real control of the experiment: they must be added to allow to assess the effect of endogenous TGFB1 on cell migration.
4. Fig. 4: what happens to p21 in the MDA-MB-231 treated with rhTGFB1?
5. Fig. 6: it is not clear to me what happens to the MDA-MB-231 in the same experimental conditions.
Author Response
Dear Editor, dear Carel:
This letter of submission is accompanied by our revised manuscript entitled:
“Autocrine TGFβ1 opposes exogenous TGFβ1-induced cell migration and growth arrest through sustainment of a feed-forward loop involving MEK-ERK signaling”
We are indebted to the reviewers for their valuable comments and suggestions and have incorporated almost all of these into the revised version of our manuscript (highlighted in the “track changes” mode). We believe that the reviewers’ critiques have substantially improved the quality of our manuscript and we are looking forward to its final acceptance in Cancers.
Sincerely yours,
Hendrik Ungefroren
Reviewer 1
Ungefroren et al described the existence of an autocrine-backed feed-forward loop involving MEK-ERK opposing exogenous TGFβ1. Although performed exclusively in vitro, the work is well designed. However, some points need to be improved.
- First of all, Results and Discussion must be better characterized. In particular, the Results section should be simplified by moving into Discussion comments involving literature data, that often make it difficult understanding the results shown.
Response: As requested, we have rephrased the introductory paragraphs of the Results subsections 2.1.-2.5. In section 2.6., we feel that the first three sentences should remain as they provide important pieces of information that need to be combined in order to rationalize the following experiments.
- Fig. 1C: it is not clear why the last band on the right was added: is it part of a different experiment? In a representative WB the bands should belong to treated and untreated samples from the same experiment and migrated to the same gel.
Response: The last band on the right in panel 1C is from the same blot, but the vertical line between lanes 3 and 4 of the left blot indicates that irrelevant lanes have been removed (as clearly stated in the figure legend). This is also evident from the uncropped blot contained in the Supplementary Material.
- Fig 2: the Ctrl siRNA and TGFB1 siRNA conditions constitute the real control of the experiment: they must be added to allow to assess the effect of endogenous TGFB1 on cell migration.
Response: As requested, the migration curves of the non-rhTGFb1 treated control cells have been included in the graphs. This issue was also raised by reviewer 4.
- Fig. 4: what happens to p21 in the MDA-MB-231 treated with rhTGFB1?
Response: We see upregulation of p21 after 8 and 24 h treatment with rhTGFb1 in immunoblots. A representative blot has been included in the Supplementary Material as Figure S4.
- Fig. 6: it is not clear to me what happens to the MDA-MB-231 in the same experimental conditions.
Response: This issue was also raised by reviewer 4. We have tested the effects of SB431542 on MDA-MB-231 cells with respect to TGFβ1 secretion and cell migration. These data have been included in the Supplementary Material as Figure S6A+B. The effects of SB431542 on MDA-MB-231 cell proliferation were published before by another group (Koo et al. 2015, Ref. #30 in the revised manuscript).
Reviewer 2 Report
Ungefroren and colleagues investigated the impact of recombinant TGFβ1 on cell growth and migration in the presence and upon depletion of autocrine TGFβ1, in two highly invasive pancreatic and breast carcinoma cell models. The data collected in this study support that the secretion of autocrine TGFβ1 has a suppressive effect on the chemokinetic activity of the cell models used and on the expression of genes associated with EMT, while treatment with recombinant TGFβ1 induces the opposite effect. Regarding cell growth, the roles of autocrine and recombinant TGB are reversed: while recombinant TGFβ1 induces growth arrest, autocrine TGFβ1 seems to have a pró-mitotic effect. The authors further suggest that autocrine TGFβ1, besides being stimulated by recombinant TGFβ1, forms a regulatory feedforward loop with ERK1/2 to sustain high-level ERK activation, which by its turn induces autocrine TGFβ1 expression (thus opposing the action of recombinant TGFβ1). Finally, the authors suggest that the ability of autocrine TGFβ1 to induce cell proliferation and inhibit cell migration is driven by ALK5-mediated induction of MEK-ERK signaling. This is a well written and interesting study; experiments are well designed, and the results of the research are mostly presented properly. Still, there are several points that the authors should clarify, as described below:
1: Despite the authors have validated siRNA KD of endogenous TGFβ1 by assessing total TGFβ1 levels in cell’s conditioned media, the measurement of the overall levels of TGFβ1 (achieved upon rhTGFβ1 treatment together with aTGFβ1 KD) is recommended. This may allow to assess how the total levels of TGFb1 (to which the cells are subjected) may account to the observed effects. This may be particularly relevant when addressing the effect of ectopic overexpression of TGFβ1.
2: Regarding the opposite effects observed with ectopic overexpression of TGFβ1 in the two cell models used, treatment of these cell models with conditioned medium containing substantial levels of TGFβ1 may help clarifying this issue.
3: Figure 4, panel B - The immunoblot shown (analysis of p21WAF1 levels) is not representative of the densitometric quantification presented below. This is particularly relevant when comparing first and third lanes.
4: Figure 5, panel B – Addressing the effect of rhTGFβ1 upon U0126-induced inhibition of cell growth is not relevant and is out of scope of this experiment. Like MDA data, these data are better as supplementary. Also, rhTGFβ1+ should be removed from Supplementary Figure S3 captions since this was not addressed.
5: Figure 6C- The reason for testing PP2 on Panc1 chemokinetic activity is unclear. This should be explained in the text for the sake of clarity.
6: Figure 6E – Similar to that performed when addressing the impact of SB431542 on cell migration (panel C), it would be more informative to evaluate SB431542 impact on cell growth in the presence and upon depletion of autocrine TGFβ1, than in the presence of rhTGFβ1.
7: Supplementary Figure S3: Figure legend should be revised: although the impact of MEK inhibitor is observed in ERK but not on SMAD3 activation, no activation of ERK is observed upon TGFβ1 treatment, as suggested by the following statement :“Activation of ERK1/2 but not of SMAD3C in response to TGFb1(...)” . Also, the use of NSC23766 as negative control should be explained.
Minor:
Line 197: clarify if it refers to the recombinant TGFβ1.
Line 197: the term “intriguingly” is misleading in this context.
Author Response
Dear Editor, dear Carel:
This letter of submission is accompanied by our revised manuscript entitled:
“Autocrine TGFβ1 opposes exogenous TGFβ1-induced cell migration and growth arrest through sustainment of a feed-forward loop involving MEK-ERK signaling”
We are indebted to the reviewers for their valuable comments and suggestions and have incorporated almost all of these into the revised version of our manuscript (highlighted in the “track changes” mode). We believe that the reviewers’ critiques have substantially improved the quality of our manuscript and we are looking forward to its final acceptance in Cancers.
Sincerely yours,
Hendrik Ungefroren
Reviewer 2
Ungefroren and colleagues investigated the impact of recombinant TGFβ1 on cell growth and migration in the presence and upon depletion of autocrine TGFβ1, in two highly invasive pancreatic and breast carcinoma cell models. The data collected in this study support that the secretion of autocrine TGFβ1 has a suppressive effect on the chemokinetic activity of the cell models used and on the expression of genes associated with EMT, while treatment with recombinant TGFβ1 induces the opposite effect. Regarding cell growth, the roles of autocrine and recombinant TGB are reversed: while recombinant TGFβ1 induces growth arrest, autocrine TGFβ1 seems to have a pro-mitotic effect. The authors further suggest that autocrine TGFβ1, besides being stimulated by recombinant TGFβ1, forms a regulatory feedforward loop with ERK1/2 to sustain high-level ERK activation, which by its turn induces autocrine TGFβ1 expression (thus opposing the action of recombinant TGFβ1). Finally, the authors suggest that the ability of autocrine TGFβ1 to induce cell proliferation and inhibit cell migration is driven by ALK5-mediated induction of MEK-ERK signaling. This is a well written and interesting study; experiments are well designed, and the results of the research are mostly presented properly. Still, there are several points that the authors should clarify, as described below:
1: Despite the authors have validated siRNA KD of endogenous TGFβ1 by assessing total TGFβ1 levels in cell’s conditioned media, the measurement of the overall levels of TGFβ1 (achieved upon rhTGFβ1 treatment together with aTGFβ1 KD) is recommended. This may allow to assess how the total levels of TGFb1 (to which the cells are subjected) may account to the observed effects. This may be particularly relevant when addressing the effect of ectopic overexpression of TGFβ1.
Response: We agree with the reviewer and in order to assess the overall levels of TGFβ1 in culture supernatants have performed ELISA-based measurements of ectopically produced total TGFβ1 in combination with added rhTGFβ1. The following cultures were tested: 1) Panc1- and MDA-MB-231 empty vector control cells and Panc1- and MDA-MB-231 cells with ectopic TGFβ1 expression, each with or without added rhTGFβ1 (5 ng/ml). The data have been included in the Supplementary Material as Figure S3. They show the expected amounts and correlate well with the respective migration curves.
2: Regarding the opposite effects observed with ectopic overexpression of TGFβ1 in the two cell models used, treatment of these cell models with conditioned medium containing substantial levels of TGFβ1 may help clarifying this issue.
Response: We agree with this suggestion, however, this might also expose the cells to a variety of potential promigratory factors other than TGFβ1, which could further complicate interpretation of the migration data.
3: Figure 4, panel B - The immunoblot shown (analysis of p21WAF1 levels) is not representative of the densitometric quantification presented below. This is particularly relevant when comparing first and third lanes.
Response: This blot has been replaced by a better one. In addition, we have added corresponding data for MDA-MB-231 cells to Fig. 4. We should mention, however, that the graph depicts the mean densitometric values from three blots (derived from three separate experiments). As a consequence, the mean values may slightly differ from those of the specific blot shown.
4: Figure 5, panel B – Addressing the effect of rhTGFβ1 upon U0126-induced inhibition of cell growth is not relevant and is out of scope of this experiment. Like MDA data, these data are better as supplementary. Also, rhTGFβ1+ should be removed from Supplementary Figure S3 captions since this was not addressed.
Response: The data with rhTGFβ1 have been deleted from the manuscript. Equivalent data for MDA-MB-231 cells have been added to the manuscript and are shown in the new Figure S7. The caption of Figure S3 (now Figure S5 in the revised manuscript) has been rephrased.
5: Figure 6C- The reason for testing PP2 on Panc1 chemokinetic activity is unclear. This should be explained in the text for the sake of clarity.
Response: As requested, the reason has been explained in section 2.6. We have previously shown that PP2 blocks TGFβ/Smad and p38 MAPK signaling in a Src-unrelated fashion (Ref. #41 in the revised version).
6: Figure 6E – Similar to that performed when addressing the impact of SB431542 on cell migration (panel C), it would be more informative to evaluate SB431542 impact on cell growth in the presence and upon depletion of autocrine TGFβ1, than in the presence of rhTGFβ1.
Response: The rhTGFβ1 was only used as a positive control for the action of SB431542! As suggested, this experiment has been performed in TGFB1-KD cells and we observed that the combination of SB431542 and TGFB1 KD provided only little more effect than each compound alone, reinforcing our model that ALK5-mediated ERK activation and aTGFb1 are part of the same pathway. In response to a request from Reviewer 4, we have also included in Figure 6E data with the p38 MAPK inhibitor SB203580 as a negative control.
7: Supplementary Figure S3: Figure legend should be revised: although the impact of MEK inhibitor is observed in ERK but not on SMAD3 activation, no activation of ERK is observed upon TGFβ1 treatment, as suggested by the following statement :“Activation of ERK1/2 but not of SMAD3C in response to TGFb1(...)” . Also, the use of NSC23766 as negative control should be explained.
Response: The reviewer is correct. Both issues have been rectified in Figure S5 (formerly Figure S3). See also point 4. The use of NSC23766 as a negative control is explained in the legend to Figure S5 (based on data from the new Ref. #58). In response to a request from Reviewer 4, the original publication on this compound has been added to the reference list (#57).
Line 197: clarify if it refers to the recombinant TGFβ1.
Response: This sentence has been rephrased to enhance clarity.
Line 197: the term “intriguingly” is misleading in this context.
Response: This term has been replaced with “Of note”.
Reviewer 3 Report
The manuscript entitled "Autocrine TGFβ1 opposes cellular mi-2 induced by exogenous TGFβ1 gration and stunting by maintaining a feed-for-3 ward cycle involving MEK-ERK signaling” is interesting and well structured.
The experiments conducted support the authors' thesis. The literature used is adequate to support both the article and the results shown.
However, I have some small suggestions for the authors:
1) the figures must be redone, the resolution is poor.
2) in both introduction and discussion, the authors never referred to how early diagnosis would be helpful in both pancreatic cancer and triple negative breast cancer (doi: 10.3390 / genes11010014; doi: 10.1016 / j.bpg .2010.02.007; DOI: 10.5306/wjco.v5.i3.283).
3) Figure 4B is not acceptable. GAPDH has some burrs.
Use another image or repeat the western.
Author Response
Dear Editor, dear Carel:
This letter of submission is accompanied by our revised manuscript entitled:
“Autocrine TGFβ1 opposes exogenous TGFβ1-induced cell migration and growth arrest through sustainment of a feed-forward loop involving MEK-ERK signaling”
We are indebted to the reviewers for their valuable comments and suggestions and have incorporated almost all of these into the revised version of our manuscript (highlighted in the “track changes” mode). We believe that the reviewers’ critiques have substantially improved the quality of our manuscript and we are looking forward to its final acceptance in Cancers.
Sincerely yours,
Hendrik Ungefroren
Reviewer 3
The manuscript entitled "Autocrine TGFβ1 opposes exogenous TGFβ1-induced cell migration and growth arrest through sustainment of a feed-forward cycle involving MEK-ERK signaling” is interesting and well structured. The experiments conducted support the authors' thesis. The literature used is adequate to support both the article and the results shown. However, I have some small suggestions for the authors:
1) the figures must be redone, the resolution is poor.
Response: We apologize for this, but the resolution is exactly the same as in our previous publications in this journal. We assume that this only applies to the copy of the PDF version.
2) in both introduction and discussion, the authors never referred to how early diagnosis would be helpful in both pancreatic cancer and triple negative breast cancer (doi: 10.3390 / genes11010014; doi: 10.1016 / j.bpg .2010.02.007; DOI: 10.5306/wjco.v5.i3.283).
Response: As requested, we have added a sentence on this issue in the “Introduction” along with two of the three suggested references (Refs #3 and 4 in the revised version).
3) Figure 4B is not acceptable. GAPDH has some burrs. Use another image or repeat the western.
Response: This concern was also raised by the other reviewers. We have replaced this blot with another one of better quality.
Reviewer 4 Report
In this manuscript, Ungefroren et al. follow up on their recently reported and unanticipated observation that endogenously expressed TGFβ1 inhibits cell motility in cells lines with high levels of autocrine TGFβ1 (aTGFβ1) and mutant KRAS (e.g. PDAC and MDA-MB-231 cells). Specifically, the authors tested if aTGFβ1 were able to antagonize the ability of exogenously applied recombinant human TGFβ1 (rhTGFβ1) to promote cell motility and growth arrest in either PDAC or MDA-MB-231 cells. Ungefroren et al. revealed that the siRNA-mediated down regulation of endogenous TGFβ1 resulted in the sensitization of genes important for the epithelial-to-mesenchymal transition and cell motility to up-regulation by rhTGFβ1. They also showed that TGFβ1 over-expression decreased the basal and rhTGFβ1-induced motility in MDA-MB-231 cells, while Panc1 cells exhibited the opposite response. Moreover, Ungefroren et al. demonstrated that TGFβ1-depletion reduced basal proliferation and stimulated the growth inhibition by rhTGFβ1 and induced the expression of p21WAF1. Furthermore, the authors report that aTGFβ1 promotes MEK-ERK signaling and vice versa resulting in the formation of a feed-forward loop, which is sensitive to an inhibitor of the TGFβ type 1 receptor ALK5 (SN431542). Taken together, Ungefroren et al. propose that in transformed cells an ALK5-MEK-ERK-aTGFβ1 pathway counteracts the promigratory and growth-arresting function of rhTGFβ1. Overall, this is a relatively well-written manuscript that addresses a question that has critical implications on the role of TGFβ signaling in cancer as well as the development and use of TGFβ inhibitors as cancer therapeutics. That being said, I feel that there are several major and minor issues that the authors need to address before I am able to recommend their manuscript for publication. These issues are described below.
Major issues:
- My biggest criticism of this work is its significant lack of single-cell imaging-based experiments. The population level experiments performed here completely ignore important single-cell heterogeneities, which have important implications for the analysis of their results. This is particularly true for the xCELLigence®-based experiments reported in Figures 2, 3A, 3B, 3C, 6C, and 6D).
- The authors repeatedly state that they were surprised to find that TGFβ1-depletion in PDAC and TNBC-derived cancer cells with known aTGFβ1 production allowed exogenous TGFβ1 to elicit a more pronounced migratory and growth-inhibitory response. However, I cannot see how this is a surprising finding, as aTGFβ1 and rhTGFβ1 are presumably both competing for the same TGFβ receptors present on the cell surface. Thus, in the absence of endogenous TGFβ1 the exogenously applied rhTGFβ1 would be unimpeded in their access to the aforementioned TGFβ receptors. Am I missing something?
- An ELISA was used to test the efficiency of the siRNA-mediated knockdown of TGFβ in the Panc1 and MDA-MB-231 cell lines. Since the ELISA was used to measure the amount of TGFβ secreted by the cells into the media, I would like to see a Western blot of cell lysates from Panc1 and MDA-MB-231 cells that were treated with the control or TGFβ-targeting siRNA to determine the amount of TGFβ that is still present within these cells.
- Rescue of TGFβ siRNA-induced phenotypes by re-expressing siRNA-resistant TGFβ in cells treated with the TGFβ-targeting siRNA. This is gold standard for controlling for off-target effects in RNAi experiments. Similar criticisms could be made for the Rac1B or SMAD2 siRNA-mediated depletion experiments presented in Figures 3C and 6F, respectively.
- Figure 1:
- Why was the expression of SERPINE1 and F2RL1 assessed in the Panc1 cells but not the MDA-MB-231 cells?
- What was the antibody that was used to probe for Snail in the Western blot provided in Figure 1C? Does it recognize both Snail1 and Snail2?
- Figure 2:
- It would be good if the authors included plots of their Panc1 and MDA-MB-231 treated with control or TGFβ siRNA but not stimulated with rhTGFβ1. These data would provide important baselines for comparison.
- The authors state in the title for this figure that they are reporting the “Effect of ectopic overexpression of TGFβ1 in MDA-MB-231 and Panc1 cells on random cell motility”. However, I would like to point out that in the Materials and Methods the authors note that they coated the lower side of the membrane of the CIM plate with a mixture of collagens I and IV. Thus, it is highly likely that what the authors are measuring is a durotaxis/invasion response and not a “random cell migration”, as described. This criticism also applies to the other experiments performed using the xCELLigence® system (i.e. Figures 3A, 3B, 3C, 6C, and 6D).
- I would like to see some images of the cells from the E-Plate View Analysis measured in these experiments. It would be good to know if they were different in their morphologies in any way. This criticism also applies to the other experiments performed using the xCELLigence® system (i.e. Figures 3A, 3B, 3C, 6C, and 6D).
- Figure 3:
- Why did the authors only test the effect of TGFβ1 over-expression on the expression of SNAI1and SNAI2 in Pnac cells (Fig. 3D)? I would like to see the same analysis for the MDA-MB-231 cells.
- In the legend for Figure 3A, the authors state “Differences between curve B and curve D were first significant at 3:15 and all later time points”. In the legend for Figure 3B, the authors state “Differences between curve B and curve D were first significant at 7:30 and all later time points”. In the legend for Figure 3C, the authors state “Differences between curve B and curve D were first significant at 1:00 and at all later time points”. The authors should somehow indicate significance in Figures 3A-C as well as indicate how this significance was determined.
- Figure 4:
- Why were the MDA-MB-231 cells not subjected to the same analysis as the Panc1 cells were regarding the levels of p21WAF1 in Figure 4B?
- The quality of the Western blot of p21WAF1 presented in Figure 4B is concerning, as it does not reflect what is shown in the plot beneath it. More specifically, the amount of p21WAF1 in the rhTGFβ1/C-siRNA lane does not look much greater than the amount in the C-siRNA alone lane, which differs from the quantification presented in the plot below the blot. This may be related to the wavy GAPDH band in the rhTGFβ1/C-siRNA lane of the blot.
- Figure 5:
- Given that the authors use an ELISA to measure the amount of TGFβ1 secreted by cells into their growth media to validate their TGFβ1-targeting siRNAs, why are they using qPCR to measure the expression of TGFβ1 mRNA in panel A? I would like to see them either use the ELISA approach or Western blotting for these experiments, as the levels of mRNA do not always correlate with protein levels in cells.
- It would be good for the authors to also include a Western blot for the levels of total ERK1/2 in Figure 5C. This is an important control.
- Figure 6:
- I would like the authors to actually show their SB203580 data, as I believe that they provide an important control for their inhibitor experiments.
- Why did the authors not test the effects of SN431542 on endogenous TGFβ1 expression, ERK activation, and cell migration in the MDA-MB-231 cells?
Minor issues:
- Lines 98-100: Since the authors note in the Materials and Methods that their cell growth media contained 10% fetal bovine serum (FBS), I find it strange that the authors make the statement “This is a serious issue since in most studies experiments were performed in medium with 10% fetal bovine serum (FBS), which may have contained high concentrations of latent or bioactive TGFβ1”. Are the authors using a special kind of FBS that lacks “high concentrations of latent or bioactive TGFβ1”?
- The positions and sizes of the molecular weight markers of the standards used in the Western blots provided in Figures 1C, 3C, 4B, 5C, 6B, and S2 need to be indicated.
- Be consistent with how you report significance in your plots. For example the plots shown in Figure 1C.
- Line 76: Delete the “can” found before “also”.
- Lines 241-243: The authors need to provide references for the various inhibitors used in this work.
- Lines 357-358: The authors need to provide a reference for their statement regarding SMAD7.
- Line 419: The word “derived” should be changed to “arrived”.
- Line 536: The “CIM plate-16” needs to be described.
Author Response
Dear Editor, dear Carel:
This letter of submission is accompanied by our revised manuscript entitled:
“Autocrine TGFβ1 opposes exogenous TGFβ1-induced cell migration and growth arrest through sustainment of a feed-forward loop involving MEK-ERK signaling”
We are indebted to the reviewers for their valuable comments and suggestions and have incorporated almost all of these into the revised version of our manuscript (highlighted in the “track changes” mode). We believe that the reviewers’ critiques have substantially improved the quality of our manuscript and we are looking forward to its final acceptance in Cancers.
Sincerely yours,
Hendrik Ungefroren
Reviewer 4
In this manuscript, Ungefroren et al. follow up on their recently reported and unanticipated observation that endogenously expressed TGFβ1 inhibits cell motility in cells lines with high levels of autocrine TGFβ1 (aTGFβ1) and mutant KRAS (e.g. PDAC and MDA-MB-231 cells). Specifically, the authors tested if aTGFβ1 were able to antagonize the ability of exogenously applied recombinant human TGFβ1 (rhTGFβ1) to promote cell motility and growth arrest in either PDAC or MDA-MB-231 cells. Ungefroren et al. revealed that the siRNA-mediated down regulation of endogenous TGFβ1 resulted in the sensitization of genes important for the epithelial-to-mesenchymal transition and cell motility to up-regulation by rhTGFβ1. They also showed that TGFβ1 over-expression decreased the basal and rhTGFβ1-induced motility in MDA-MB-231 cells, while Panc1 cells exhibited the opposite response. Moreover, Ungefroren et al. demonstrated that TGFβ1-depletion reduced basal proliferation and stimulated the growth inhibition by rhTGFβ1 and induced the expression of p21WAF1. Furthermore, the authors report that aTGFβ1 promotes MEK-ERK signaling and vice versa resulting in the formation of a feed-forward loop, which is sensitive to an inhibitor of the TGFβ type 1 receptor ALK5 (SN431542). Taken together, Ungefroren et al. propose that in transformed cells an ALK5-MEK-ERK-aTGFβ1 pathway counteracts the promigratory and growth-arresting function of rhTGFβ1. Overall, this is a relatively well-written manuscript that addresses a question that has critical implications on the role of TGFβ signaling in cancer as well as the development and use of TGFβ inhibitors as cancer therapeutics. That being said, I feel that there are several major and minor issues that the authors need to address before I am able to recommend their manuscript for publication. These issues are described below.
Major issues:
- My biggest criticism of this work is its significant lack of single-cell imaging-based experiments. The population level experiments performed here completely ignore important single-cell heterogeneities, which have important implications for the analysis of their results. This is particularly true for the xCELLigence®-based experiments reported in Figures 2, 3A, 3B, 3C, 6C, and 6D).
Response: We agree with the Reviewer, however, the xCELLigence technology is designed for analyzing populations of cells and is not suitable for single-cell analyses. We should say that our study includes already two cell types, two cellular responses, and several different target genes of TGFB1. Moreover, the population of Panc1 cells (and likely also MDA-MB-231 cells) is known to be heterogeneous comprising cells of different EMT phenotypes at any time. We feel that analysing all this at the single cell level and heterogeneities within a cell line is beyond the scope of this work. However, we have monitored morphological changes and found that in TGFB1-KD cells rhTGFβ1 was able to induce a larger number of spindle-shaped cells than in controls. These data have been added to the manuscript and are presented in the new Figure S1. We sincerely hope that this will satisfy the concerns of the Reviewer.
- The authors repeatedly state that they were surprised to find that TGFβ1-depletion in PDAC and TNBC-derived cancer cells with known aTGFβ1 production allowed exogenous TGFβ1 to elicit a more pronounced migratory and growth-inhibitory response. However, I cannot see how this is a surprising finding, as aTGFβ1 and rhTGFβ1 are presumably both competing for the same TGFβ receptors present on the cell surface. Thus, in the absence of endogenous TGFβ1 the exogenously applied rhTGFβ1 would be unimpeded in their access to the aforementioned TGFβ receptors. Am I missing something?
Response: Our finding that aTGFβ1 inhibits cell motility and promotes proliferation is completely novel. The vast majority of publications report on autocrine TGFβ1 being pro-invasive or antiproliferative, thus acting in the same way as exogenous/rh TGFβ1. Moreover, there are no studies, to the best of our knowledge, which have analysed how depletion of endogenous/aTGFβ1 impacts the migratory/antiproliferative response to rhTGFβ1.
An ELISA was used to test the efficiency of the siRNA-mediated knockdown of TGFβ in the Panc1 and MDA-MB-231 cell lines. Since the ELISA was used to measure the amount of TGFβ secreted by the cells into the media, I would like to see a Western blot of cell lysates from Panc1 and MDA-MB-231 cells that were treated with the control or TGFβ-targeting siRNA to determine the amount of TGFβ that is still present within these cells.
Response: Successful downregulation of both TGFβ1 mRNA (by qPCR) and protein (by Western blotting of cell lysates) following TGFB1 siRNA transfection has extensively been analyzed in a previous publication (#16 in the reference list of the revised manuscript, see Figures S1B, S5A and S7 in this paper). We believe that the ELISA data of culture supernatants are more relevant here as secreted TGFβ1 represents the biologically active form, which mediates the antimigratory function (see Ref. #16).
- Rescue of TGFβ siRNA-induced phenotypes by re-expressing siRNA-resistant TGFβ in cells treated with the TGFβ-targeting siRNA. This is gold standard for controlling for off-target effects in RNAi experiments. Similar criticisms could be made for the Rac1B or SMAD2 siRNA-mediated depletion experiments presented in Figures 3C and 6F, respectively.
Response: All the siRNAs used in this study were pre-evaluated by the supplying companies and guaranteed to lack off-target effects. We therefore believe that re-expressing siRNA-resistant TGFβ in cells, although being admittedly a smart approach, is not mandatory here. Moreover, as shown in Figure 3, ectopically expressed TGFb1 can either behave like aTGFβ1 and inhibit migration (in MDA-MB-231 cells), or like rhTGFb1 to stimulate migration (in Panc1 cells). It is quite likely that re-expression of an siRNA-resistant TGFβ in cells will exhibit the same different behavior, which might complicate interpretation of this control experiment.
- Figure 1:
- Why was the expression of SERPINE1 and F2RL1 assessed in the Panc1 cells but not the MDA-MB-231 cells?
Response: We have now assessed the response of both genes in MDA-MB-231TGFB1-KD cells and found F2RL1/PAR2 but not PAI-1 expression to be enhanced by rhTGFb1. The data for F2RL1 have been added to Figure 1B.
- What was the antibody that was used to probe for Snail in the Western blot provided in Figure 1C? Does it recognize both Snail1 and Snail2?
Response: We apologize for not having listed the Snail1 antibody in the Material and Methods section. This has been rectified in the revised version. This antibody is specific for Snail1.
- Figure 2:
- It would be good if the authors included plots of their Panc1 and MDA-MB-231 treated with control or TGFβ siRNA but not stimulated with rhTGFβ1. These data would provide important baselines for comparison.
Response: As requested, the corresponding curves have been included in Figure 2. This issue was also raised by reviewer 1.
- The authors state in the title for this figure that they are reporting the “Effect of ectopic overexpression of TGFβ1 in MDA-MB-231 and Panc1 cells on random cell motility”. However, I would like to point out that in the Materials and Methods the authors note that they coated the lower side of the membrane of the CIM plate with a mixture of collagens I and IV. Thus, it is highly likely that what the authors are measuring is a durotaxis/invasion response and not a “random cell migration”, as described. This criticism also applies to the other experiments performed using the xCELLigence® system (i.e. Figures 3A, 3B, 3C, 6C, and 6D).
Response: It is correct that only the lower side of the CIM plate membrane is coated with the collagens. From the architecture of the CIM plate (for reference see https://www.agilent.com/en/product/cell-analysis/real-time-cell-analysis/rtca-microplates/rtca-cim-plates-741221#productdetails) it should become clear that the cells can only come in contact with the collagens once they have managed to migrate through the pores to reach the underside or, in other words, the cells have already passed through the pores to reach the lower chamber before they have the chance to come in contact with the collagens. The collagen coating serves to keep the cells attached to the underside of the membrane to remain in contact with the electrodes as long as possible to enhance signal strength. For an invasion assay, Matrigel is layered on top of the upperside of the membrane (in the upper chamber). The invasion setup has been used by us previously (see PMID: 23457587, PMID: 22699812).
- I would like to see some images of the cells from the E-Plate View Analysis measured in these experiments. It would be good to know if they were different in their morphologies in any way. This criticism also applies to the other experiments performed using the xCELLigence® system (i.e. Figures 3A, 3B, 3C, 6C, and 6D).
Response: This is a good idea. However, we were not using E-plates-View, meaning that we cannot simply take microphotographs of the cells, because in the CIM plates the view is blocked by the semipermeable membrane separating the upper from the lower chamber. As outlined above and in accordance with the higher migratory activity of TGFB1-KD vs. control cells upon rhTGFβ1 treatment, we observed a higher percentage of spindle-shaped cells in these cultures (see Figure S1).
- Figure 3:
- Why did the authors only test the effect of TGFβ1 over-expression on the expression of SNAI1and SNAI2 in Pnac cells (Fig. 3D)? I would like to see the same analysis for the MDA-MB-231 cells.
Response: We repeated these experiments with MDA-MB-231 cells and found that over-expression of TGFB1 in MDA-MB-231 cells failed to induce a statistically significant increase in SNAI1 or SNAI2 expression. These data, which have been included in Figure 3D, match the failure of MDA-MB-231 cells to respond to ectopic TGFβ1 over-expression with an increase in migratory activity.
- In the legend for Figure 3A, the authors state “Differences between curve B and curve D were first significant at 3:15 and all later time points”. In the legend for Figure 3B, the authors state “Differences between curve B and curve D were first significant at 7:30 and all later time points”. In the legend for Figure 3C, the authors state “Differences between curve B and curve D were first significant at 1:00 and at all later time points”. The authors should somehow indicate significance in Figures 3A-C as well as indicate how this significance was determined.
Response: As requested, we have indicated in all migration plots the times points at which data were first significant with an asterisk (*). Statistical significance was determined with the unpaired two-tailed Student’s t-test from the raw data (triplicate or quadruplicate wells per time point) of the various curves. The raw data are recorded during the assay and can be displayed in Excel format and subjected to statistical analysis. This explanation is now given in the legend to Figure 2 because a statistical analysis of migration data was applied here for the first time in the manuscript.
- Figure 4:
- Why were the MDA-MB-231 cells not subjected to the same analysis as the Panc1 cells were regarding the levels of p21WAF1 in Figure 4B?
Response: We did now subject MDA-MB-231 cells to the same analysis and like for Panc1 cells noted enhanced upregulation of p21WAF1 in response to a TGFB1 knockdown. We have included these data in Figure 4B.
- The quality of the Western blot of p21WAF1 presented in Figure 4B is concerning, as it does not reflect what is shown in the plot beneath it. More specifically, the amount of p21WAF1 in the rhTGFβ1/C-siRNA lane does not look much greater than the amount in the C-siRNA alone lane, which differs from the quantification presented in the plot below the blot. This may be related to the wavy GAPDH band in the rhTGFβ1/C-siRNA lane of the blot.
Response: We agree with the reviewer and have replaced this blot with one of better quality. We should mention, however, that the graph depicts the mean densitometric values from three blots (derived from three separate experiments). As a consequence, the mean values may slightly differ from those of the specific blot shown.
- Figure 5:
- Given that the authors use an ELISA to measure the amount of TGFβ1 secreted by cells into their growth media to validate their TGFβ1-targeting siRNAs, why are they using qPCR to measure the expression of TGFβ1 mRNA in panel A? I would like to see them either use the ELISA approach or Western blotting for these experiments, as the levels of mRNA do not always correlate with protein levels in cells.
Response: The TGFβ1 mRNA data were included to demonstrate that regulation occurs at the transcriptional / mRNA level. However, as requested, we have now performed, in addition, ELISAs and see the same effect, confirming that the mRNA levels correlate with the respective protein levels (see new panel B in Figure 5). The correlation between mRNA and secreted protein following TGFB1 knockdown has also been confirmed in a previous publication (#16 in the reference list of the revised manuscript, see Figure S5A in this publication).
- It would be good for the authors to also include a Western blot for the levels of total ERK1/2 in Figure 5C. This is an important control.
Response: As requested, we have repeated this blot using an antibody to ERK1/2 to control for equal loading. The samples with rhTGFb1 treatment have been deleted from this panel since treatment with rhTGFb1 had no statistically significant effect on either the control siRNA or the TGFB1 siRNA transfected cells at the 1-hour time point.
- Figure 6:
- I would like the authors to actually show their SB203580 data, as I believe that they provide an important control for their inhibitor experiments.
Response: As requested, we have included the SB203580 data in Figure 6, panel E. In response to a request from Reviewer 2, we have evaluated the impact of SB431542 on cell growth in the presence and upon depletion of autocrine TGFβ1. The data with rhTGFb1 from panel E have been deleted from the manuscript (see comment #? From Reviewer ?).
- Why did the authors not test the effects of SN431542 on endogenous TGFβ1 expression, ERK activation, and cell migration in the MDA-MB-231 cells?
Response: This issue was also raised by reviewer 1. As requested, we have tested the effects of SB431542 on MDA-MB-231 cells with respect to TGFβ1 secretion and cell migration. All these data were included in the new Figure S8. The effect of SB431542 on MDA-MB-231 cell proliferation was published before by another group (Koo et al. 2015, see Ref. #30).
Minor issues:
- Lines 98-100: Since the authors note in the Materials and Methods that their cell growth media contained 10% fetal bovine serum (FBS), I find it strange that the authors make the statement “This is a serious issue since in most studies experiments were performed in medium with 10% fetal bovine serum (FBS), which may have contained high concentrations of latent or bioactive TGFβ1”. Are the authors using a special kind of FBS that lacks “high concentrations of latent or bioactive TGFβ1”?
Response: We apologize for this confusion. In contrast to standard culture growth medium which contained 10% FBS, all migration assays were carried out in medium containing only 1% FBS and the ELISAs of TGFb1 in medium with 0.5% FBS. Regarding the migration assays, this important piece of information is contained in the Methods sections in Refs. 14, 15 and 27, which provide a more detailed description of the xCELLigence assays. Reducing the FBS concentration by 10- or 20-fold brings the exogenous TGFβ1 in the serum down to very low levels.
- The positions and sizes of the molecular weight markers of the standards used in the Western blots provided in Figures 1C, 3C, 4B, 5C, 6B, and S2 need to be indicated.
Response: The positions and sizes of molecular weight markers are shown in the uncropped blots that we had submitted as Supplementary Material along with the original submission.
- Be consistent with how you report significance in your plots. For example the plots shown in Figure 1C.
Response: The style of indicating significance in the E-cadherin blot has been changed to match that in the Snail blot.
- Line 76: Delete the “can” found before “also”.
Response: Done.
- Lines 241-243: The authors need to provide references for the various inhibitors used in this work.
Response: As requested, appropriate references have been added for U0126, SB431542, SB203580 and NSC23766 (Refs #31, 39, 42 and 57, respectively).
- Lines 357-358: The authors need to provide a reference for their statement regarding SMAD7.
Response: As requested, a reference has been added (Ref. #46).
- Line 419: The word “derived” should be changed to “arrived”.
Response: Done.
- Line 536: The “CIM plate-16” needs to be described.
Response: As requested, we have provided an URL to the supplier’s website containing a well-illustrated description of the RTCA CIM-Plate 16. More detailed descriptions were provided in our previous publications (Refs. #16, 17, and 43).
Round 2
Reviewer 1 Report
The authors responded adequately to the requests
Author Response
We sincerely thank this reviewer for this enthusiastic comment.
Reviewer 4 Report
Overall, the authors have addressed the lion’s share of my previous concerns with their original manuscript. However, I feel that a couple of important issues remain that I would like to see the authors deal with before I can recommend their article for publication. These issues are described below:
- The images presented in the new Figure S1 need a scale bar.
- Unless the siRNAs were experimentally shown by their manufacturer to specifically lack off-target effects in MDA-MB-231 and Panc1 cells, I stand by my previous criticism that the authors should show that they could rescue of TGFβ siRNA-induced phenotypes by re-expressing siRNA-resistant TGFβ in cells treated with the TGFβ-targeting siRNA. This is gold standard for controlling for off-target effects in RNAi experiments. Similar criticisms could be made for the Rac1B or SMAD2 siRNA-mediated depletion experiments presented in Figures 3C and 6F.
- Regarding the inclusion of the positions and sizes of molecular weight markers on the Western blots presented in this manuscript, I would still like to see them included in the Western blots presented in the main figures of this work.
Author Response
Overall, the authors have addressed the lion’s share of my previous concerns with their original manuscript.
Response: We thank the reviewer for acknowledging our attempts to address as many of his concerns as possible.
However, I feel that a couple of important issues remain that I would like to see the authors deal with before I can recommend their article for publication. These issues are described below:
- The images presented in the new Figure S1 need a scale bar.
Response: As requested, scale bars have been added to each of the four micrographs.
- Unless the siRNAs were experimentally shown by their manufacturer to specifically lack off-target effects in MDA-MB-231 and Panc1 cells, I stand by my previous criticism that the authors should show that they could rescue of TGFβ siRNA-induced phenotypes by re-expressing siRNA-resistant TGFβ in cells treated with the TGFβ-targeting siRNA. This is gold standard for controlling for off-target effects in RNAi experiments. Similar criticisms could be made for the Rac1B or SMAD2 siRNA-mediated depletion experiments presented in Figures 3C and 6F.
Response:
- As outlined in our previous response, all the siRNAs used in this study were pre-evaluated by the supplying companies and guaranteed to lack off-target effects. In fact, we purchased these siRNAs (rather than using self-designed siRNAs) just because we wanted to avoid the time-consuming testing for off-target effects. We consider it highly unlikely that a specific siRNA lacks off-target effects in the particular cell line(s) used by the supplier (the identity of which we do not know) but does show off-target effects in our Panc1 and MDA-MB-231 cells.
- All siRNAs used in our study have been used successfully in earlier publications (TGFB1: PMID: 33260366; RAC1B: PMID: 33567745, PMID: 32545415, PMID: 31817656, PMID: 31434318, PMID: 31108998, PMID: 29229918, PMID: 24378395; SMAD2: PMID: 31108998, PMID: 31817656, PMID:21624123).
- We have evaluated the specificity of the TGFB1 and RAC1B siRNAs at both the mRNA and protein level and within the protein level by both ELISA and immunoblotting (PMID: 33260366), which resulted in conclusive and highly reproducible results.
- As shown in Figure 3, ectopically expressed TGFb1 can either behave like aTGFβ1 and inhibit migration (in MDA-MB-231 cells), or like rhTGFb1 to stimulate migration (in Panc1 cells). In MDA-MB-231 cells re-expression of an siRNA-resistant TGFβ1 mRNA in cells is expected to display the SAME effect as the siRNA! Hence, in this cell line, the rescue approach would not work anyway. In sum, we feel that the all three siRNAs have been sufficiently tested and evaluated to produce only specific effects.
3. Regarding the inclusion of the positions and sizes of molecular weight markers on the Western blots presented in this manuscript, I would still like to see them included in the Western blots presented in the main figures of this work.
Response: As requested, the positions of and sizes of molecular weight markers have been included in all the figure panels that contain Western blots (Figures 1C, 4B, 5D and 6B).